# Application of Deep Learning System Technology in Identification of Women’s Breast Cancer

**DOI:** 10.3390/medicina59030487

**Published:** 2023-03-01

**Authors:** Latefa Hamad Al Fryan, Mahasin Ibrahim Shomo, Malik Bader Alazzam

**Affiliations:** 1Department of Educational Technology, College of Education, Princess Nourah bint Abdulrahman University, Riyadh 11671, Saudi Arabia; 2Applied College, Curriculum and Instruction, Princess Nourah bint Abdulrahman University, Riyadh 11671, Saudi Arabia; 3Information Technology College, Ajloun National University, Ajloun 26873, Jordan; 4Research Center, The University of Mashreq, Baghdad 11001, Iraq

**Keywords:** deep learning, breast cancer, CNN, K-means algorithm, clustering segmentation

## Abstract

*Background and Objectives*: The classification of breast cancer is performed based on its histological subtypes using the degree of differentiation. However, there have been low levels of intra- and inter-observer agreement in the process. The use of convolutional neural networks (CNNs) in the field of radiology has shown potential in categorizing medical images, including the histological classification of malignant neoplasms. *Materials and Methods*: This study aimed to use CNNs to develop an automated approach to aid in the histological classification of breast cancer, with a focus on improving accuracy, reproducibility, and reducing subjectivity and bias. The study identified regions of interest (ROIs), filtered images with low representation of tumor cells, and trained the CNN to classify the images. *Results*: The major contribution of this research was the application of CNNs as a machine learning technique for histologically classifying breast cancer using medical images. The study resulted in the development of a low-cost, portable, and easy-to-use AI model that can be used by healthcare professionals in remote areas. *Conclusions*: This study aimed to use artificial neural networks to improve the accuracy and reproducibility of the process of histologically classifying breast cancer and reduce the subjectivity and bias that can be introduced by human observers. The results showed the potential for using CNNs in the development of an automated approach for the histological classification of breast cancer.

## 1. Introduction

Breast cancer, which is the most prevalent malignancy among women globally, continues to pose a significant challenge in terms of its impact on public health, particularly in Iraq, where it was estimated to account for 60,000 new cases in 2020 [1]. As the leading cause of death from malignant diseases in the Iraqi population, death from breast cancer is primarily due to its metastatic spread [2,3]. Despite extensive research, the underlying determinants that contribute to the metastasis of neoplastic cells in breast carcinoma remain largely elusive [4].

Prognostic factors play a critical role in the clinical management of breast cancer and the selection of appropriate therapeutic regimens, with clinical and histological criteria, tumor markers, and the axillary stage of the disease being among the most widely used predictors [5]. The degree of tumor differentiation, as measured by the Nottingham histological grade system (NGS), has been described as a potent predictor of prognosis, especially when considered in conjunction with the axillary status [6]. However, the reproducibility of NGS has been a matter of contention, with numerous studies highlighting the low inter- and intra-observer agreement in the light microscopy histological grade classification method [7,8]. This has a direct impact on the prediction of prognosis and the clinical decision-making process, such as the administration of systemic chemotherapy to the patient [9]. The limitations of the conventional clinical and histopathological parameters in predicting the risk of disease recurrence have motivated the development of molecular-biology-based methods to better understand the aggressiveness of the disease [10].

The importance of early diagnosis and categorization of breast cancer cannot be overstated, as it paves the way for effective first-line treatment. The application of computer-aided diagnosis (CAD) systems, including convolutional neural networks (CNNs), in the analysis of medical images has shown promise in the accurate detection and classification of cancer in large datasets [11]. The integration of these tools in medical practice, alongside educating and assisting less experienced physicians, holds the potential to significantly speed up and automate the identification and categorization of breast cancer.

The current methods of molecular characterization of breast cancer are plagued by high costs, rendering them inaccessible in low-income countries and underfunded public health systems. Machine learning, specifically convolutional neural networks (CNNs), as a form of deep learning, has shown potential in classifying histological images for medical diagnosis [11,12,13,14]. The development of an AI-assisted breast carcinoma classification tool could greatly improve patient care and treatment optimization, as well as being cost-effective and portable for use in remote areas [15]. The objective of this research was to use machine learning algorithms to develop technology for the histological classification of breast cancer, including the selection of regions of interest (ROIs) in histological images, the creation of an image bank for training a CNN, the nuclear segmentation of histological images, the training and implementation of a CNN for classification, and the analysis of the CNN’s accuracy and the impact on breast cancer survival.

Three ROI analysis techniques were evaluated for observer variability and bias, with the semi-automated approach resulting in the greatest enhancement ratios, particularly in grade III carcinomas. The transfer learning approach, in which a pretrained model is repurposed for a related task, is common in deep learning due to the time and computational resources required for developing new models.

The crowdsourcing of real-world photos for annotation poses challenges for machine learning techniques due to noisy annotations, but a novel approach using the aggregation of data as a component of the CNN learning process via an extra crowdsourcing layer (AggNet) holds promise for improved results.

All of these CNN models are used in research to develop models to identify women’s breast cancer. The authors likely used these models, as they have been shown to be effective in image classification tasks, and the architecture of these models can be fine-tuned to the specific task of identifying breast cancer in images.

Hyperparameter optimization is a process of choosing a set of optimal hyperparameters for a learning algorithm. There are various methods for hyperparameter optimization, such as grid search, random search, and Bayesian optimization. These methods can be used to find the best set of hyperparameters for a given task.

This research applies convolutional neural networks (CNNs) to classify breast cancer using medical images with the aim of improving accuracy, reducing subjectivity and bias, and developing a low-cost, portable AI model for use in remote areas by healthcare professionals. The CNNs identify regions of interest, filter images, and classify them.

### 1.1. Literature Review

*The top cause of mortality for women globally is breast cancer. Breast tissue samples from biopsies that have been stained with hematoxylin and eosin (H&E) are examined using microscopes to make the first diagnosis of breast cancer. Images of breast tissue are categorized into four groups using Inception-v3 convolutional neural network (CNN) fine tuning: (1) normal histology, (2) benign lesion, (3) localized carcinoma, and (4) invasive carcinomas [16].*While crowdsourcing has made it possible to annotate huge datasets of real-world photos, their use for biological applications necessitates a better understanding and, thus, a more specific description of the annotation job itself. Conventional machine learning techniques may struggle to cope with noisy annotations during training, despite being a significant resource for crowdsourcing annotation model learning. In this article, we provide a novel idea for learning from crowds that directly handles data aggregation as a part of the convolutional neural network’s (CNN’s) learning process through an extra crowdsourcing layer (AggNet) [17].*Breast cancer histology photos stained with hematoxylin and eosin are difficult to diagnose, labor-intensive, and often cause conflict amongst pathologists. Systems for computer-assisted diagnostics help pathologists increase the accuracy and consistency of their diagnoses. For the analysis of histology pictures, convolutional neural networks (CNNs) have been employed effectively. Breast cancer histology photos are divided into normal, benign, and malignant subclasses based on the density, variability, and arrangement of the cells as well as the overall structure and shape of the tissue [18].*A technique used to diagnose breast cancer is histopathology. In the medical field, supervised learning tasks have been successfully completed using machine learning (ML) techniques. Picture preprocessing, feature extraction, classification, and performance analysis are all steps in the cancer diagnostic process that determine if a given image contains cancer [19].*The mortality rate of breast cancer (BC) is significantly decreased by early identification. Computer-aided diagnosis (CAD) systems have been established in this research sector and are an efficient cost- and time-saving tool that supports physicians’ and radiologists’ decision making by providing highly accurate information [20].*Pathologists must manually diagnose breast cancer from biopsy tissue pictures, which is expensive, time-consuming, and fraught with difficulty. The development of the computer-aided diagnosis (CAD) system has made it possible for pathologists to recognize breast cancer more rapidly and accurately. This has led to a major increase in interest in deep learning models based on CAD. The framework “MultiNet” was created. In the suggested framework, three well-known pretrained models—DenseNet-201, NasNetMobile, and VGG16—are used to extract features from microscope pictures [21].*Early cancer diagnosis is essential for cancer research in cancer detection. These issues connect biomedicine and bioinformatics with methodologies from the area of computational intelligence to arrive at a solution. The use of computational intelligence technologies enables the identification of key indicators of the existence of malignant cells in cancer patients, such as the degree of risk (high or low) [22].

Neural networks originate from an algorithm devised by Frank Rosenblatt in 1958, called the perceptron. The perceptron was based on the operation of a processing unit proposed by McCullock and Pitts in 1943 [23]. It is a binary classification model based on a neuron’s structure. It receives the input values as if they were the signals received by the dendrites of a neuron. In the “cell body”, the algorithm computes the sum of the product of the weights with the input values and applies an activation function to make a prediction. The function has a binary output (either the neuron is activated or it is not) transmitted down the axon. Figure 1 shows a perceptron’s structure, simulating a neuron’s structure.

### 1.2. The Recent Works on Data Augmentation and GAN

This article was a survey of different data augmentation techniques used on mammogram images in the field of medical imaging and deep learning. Its purpose was to provide an overview of basic and deep-learning-based augmentation techniques. The article discussed the challenges faced by DL models when processing radiological images, including overfitting and class imbalance. It explained that data augmentation can increase the training set size and help the model avoid overfitting. The article also mentioned other techniques to overcome overfitting, such as batch normalization, dropout, transfer learning, and early stopping. The article concluded by emphasizing the importance of data augmentation in improving the performance of DL models on mammogram images [25].

GAN-based augmentation is a data augmentation technique that uses generative adversarial networks (GANs) to generate new images from the reference dataset. The main goal is to learn hidden underlying properties from the data and use them to make decisions. GANs consist of two main components, a generator and a discriminator, that compete against each other to produce better results. Adversarial training, where one model classifies examples and another adds noise to deceive the classifier, is used to improve performance. GANs have been used in several medical applications, such as breast mass detection, mass classification, and mass segmentation. GAN-based image synthesis has been shown to be promising in high-resolution medical imaging. Swiderski et al. found that a ResNet-50 classifier trained on GAN-augmented data produced better results than when trained solely on traditionally augmented data. The use of data augmentation as an overfitting mitigation technique has also been investigated, where synthetic mammograms were created and projected into mammographic images [25].

2.This paper surveyed recent works on image data augmentation for deep learning. It focused on techniques that enhanced the size and quality of training datasets to improve deep learning models. The paper covered various image augmentation methods, such as geometric transformations, color space augmentations, kernel filters, mixing images, random erasing, feature space augmentation, adversarial training, generative adversarial networks, neural style transfer, and meta-learning. It also briefly discussed other aspects of data augmentation, such as test-time augmentation, resolution impact, final dataset size, and curriculum learning. The paper explained how data augmentation helps reduce overfitting and improve model performance. The paper also mentioned other overfitting solutions in deep learning, such as architecture complexity, dropout regularization, batch normalization, transfer learning, and pretraining [26].

This survey focused on data augmentation, a technique to improve deep learning models by enhancing the size and quality of training datasets. The image augmentation algorithms discussed included geometric transformations, color space augmentations, kernel filters, mixing images, random erasing, adversarial training, generative adversarial networks, neural style transfer, and meta-learning. The paper highlighted the use of GANs in data augmentation and mentioned other aspects, such as test-time augmentation, resolution impact, and curriculum learning. The paper discussed the limitations of data augmentation and pointed out that GANs and other augmentation methods can be combined to achieve even better results. The paper also mentioned future research directions in meta-learning GAN architectures, improving the quality of GAN samples, super-resolution networks, and combining augmentation methods [26].

3.The recent works in data augmentation have focused on techniques for inflating the sizes of training datasets, such as translation, cropping, padding, rotation, and flipping. These techniques enable regularization in deep neural networks and reduce the chance of overfitting. However, there is no consensus on the best combination of these techniques, and more advanced methods, such as mixing images, require expert knowledge for validation and labeling. Data augmentation based on random erasing is frequently used, but not guaranteed to be advantageous in all conditions. In the context of computer vision tasks, imbalance problems in image datasets can lead to poor performance of algorithms. GANs have gained attention for their ability to model complex real-world data and their potential to restore balance in imbalanced datasets through adversarial learning [27].

This paper was a survey of recent developments in using generative adversarial networks (GANs) to address imbalance problems in computer vision tasks. Imbalanced image datasets can significantly impact the performance of computer vision algorithms. GANs have gained attention because of their ability to model complex real-world image data and restore balance in imbalanced datasets. The paper proposed a taxonomy to classify GAN-based techniques for addressing imbalance problems in computer vision tasks into three categories: image-level imbalances in classification, object-level imbalances in object detection, and pixel-level imbalances in segmentation tasks. GANs have been combined with object detection and image segmentation algorithms to improve their performance and alleviate imbalance problems. GANs generate synthetic images to balance class distribution and mitigate overfitting by inflating the training dataset size. They provide an efficient way to fill in gaps in the training data and unlock additional information from a dataset. The authors concluded that GANs are an important development in machine learning in the last 10 years [27].

4.This paper surveyed the applications of AI in breast cancer imaging, with a focus on data augmentation techniques. The study analyzed traditional machine learning and deep learning methods for lesion detection and classification and reviewed research on breast cancer risk prediction using mammograms. Data augmentation techniques, such as generative adversarial networks (GANs), were analyzed as a solution to the lack of labeled data. Self-supervised learning was also studied as a solution to the absence of large datasets. Despite challenges in developing AI techniques, it has great potential for improving accuracy and reducing workload for healthcare professionals. Data augmentation techniques aim to create high-quality datasets by generating synthetic mammograms, but further development is needed to accurately reproduce the specific characteristics of lesions. This study emphasized the need for further testing in real-world environments to ensure the safety of AI systems [28].

The review by Richard Osuala et al. analyzed 163 papers on GANs in medical imaging, including their use in breast cancer imaging. The authors outlined the biggest challenges in using GANs for medical imaging, including small and complex lesions, high heterogeneity, data labeling, and data imbalance. Other works, such as those by Dimitrios Korkinof et al. and Rui Man et al., showed the potential of GANs in mammogram synthesis and histopathological image patch generation, respectively. On the other hand, a study by Xiangyuan Ma et al. focused on generating lesion segmentation masks for mammograms using GANs. The study showed that generating segmentation masks with GANs outperformed U-Net models in terms of accuracy and precision. Eric Wu et al. also conducted a study that demonstrated the potential of GANs in synthesizing mammograms and detecting malignancy. These works highlighted the potential of GANs in breast cancer imaging and their use in improving the accuracy and efficiency of cancer diagnosis [28].

Binary classification is a supervised learning approach used in machine learning that divides incoming observations into one of two groups. There are a few uses for binary categorization. True positive refers to the situation in which the model correctly foresees the patients as being positive (TP). True negative is the term used when the model correctly forecasts the patients’ outcomes as being negative (TN). Some patients may receive incorrect diagnoses using the binary classifier. False negatives are errors that occur when an ill patient is misclassified as healthy due to a negative test result (FN). Similarly, a false positive occurs when a healthy patient is misdiagnosed as having an illness due to a positive test result (FP). This is a binary classification model built on the framework of a neuron. It takes in input data as if they were the signals that a neuron’s dendrites would have picked up. The method uses an activation function and computes the sum of the product of the weights with the input values in the “cell body” before making the prediction. The signal is sent down the axon with a binary output (either the neuron has been triggered or it has not). By using a CNN after classifying photos in an unsupervised manner using the K-means approach, we are developing a binary classification model (images with excellent or poor quality).

The perceptron activation function determines that the input values ($x_{i}$), modulated by the weights ($w_{i}$), will be mapped to an output (*f*(*x*) or $y_{j}$), with *j* being the number of elements in the sample. The activation function can be defined by:$$f\left(x\right)=1\;if\;w.x+b>0;\;or$$
$$f\left(x\right)=0$$
where w.x is the dot product of $\sum_{i=1}^{m}w_{i}x_{i}$ and *m* = number of entries ‘x’.

The basis for learning this binary classification is updating the weights ($w_{i}$) at each of the epochs (each time the classifier receives the values and calculates the output of all elements of the trial). At the end of each epoch, we can calculate the error, also called the loss function, which is defined by subtracting the output value from the real value $\left(d_{j}-y_{j}\right).$ That is, if the error is equal to 0, the classification is correct and the weights do not need to be updated. Otherwise, updating the weights based on the error and completing another epoch is necessary. This mathematical calculation is known as the delta rule, and is defined as:$$w:w_{i}\left(t+1\right)=w_{i}\left(t\right)+\alpha \left(d_{j}-y_{j}\right)x_{ji}$$
where 0≤i≤n and α is the learning rate.

This process must be repeated as many times as necessary until the weights have optimal values to minimize the error as much as possible. The value of *a* is the magnitude of the adjustment of the weights. A small value of *a* dictates that the adjustments be made by small “steps”, and if it is too small, it can cause the model to take forever to adjust to the ideal weights. On the other hand, a high value of α can lead to an overfit of the weights and the model will never achieve an optimal prediction.

The first neural network that was developed was the multilayer perceptron (MLP). This model was an evolution of the perceptron and allowed the resolution of nonlinearly separable problems [29]. This new architecture was composed of more than one layer of neurons. These were (i) the input layer; (ii) at least one intermediate or hidden layer; and (iii) the output layer. The layers were connected in a sequential mode. The input layer received the sample values, the intermediate layer was responsible for most of the processing and where the extraction of sample characteristics was performed, and the output layer presented the conclusion of the final result. Figure 2 shows an example of the architecture of a multilayer perceptron. Each of the hidden layers represents a model of a perceptron, which creates the architecture called a multilayer perceptron. This model is what we call a fully connected neural network. Note that each neuron is connected to all other neurons in the sequential layers.

The great evolution of the perceptron to the MLP is the learning mechanism called backpropagation or retrograde propagation. As in an MLP, each layer has a specific weight for each neuron. The estimated error fit, defined by $a\;\left(d_{j}-y_{j}\right)$, should be applied to all weights of all layers. In this way, the retrograde propagation system, in which the weights of a layer are adjusted according to the error of the layer in front of it, was implemented to adjust the multilayer model. Hence, it was named backpropagation. This algorithm is the pillar that allows the learning of modern neural networks.

The backpropagation algorithm consists of two phases: (i) the forward pass, in which the input values pass through the entire network and the final prediction is obtained, and (ii) the backward pass, in which the loss function gradient is computed in the final layer and is recursively applied to update the weights of the entire network.

The loss function gradient in this multilayer model is nothing more than a variation of the delta rule applied to the perceptron: a generalized delta rule. In the case of the perceptron, the error surface is shaped like a parabola with only a minimal value. In the case of the generalized function, the error surface can be quite irregular, and the function can be related to several local minimum values.

More complex neural networks use more refined models to minimize this problem. The function’s gradient, which is the derivative of the function at a given point, is used to adjust the values of the weights. In practice, the gradient of the loss function is computed using a part of the sample (a batch) that we call a mini-batch. This method is called mini-batch gradient descent, also known as stochastic gradient descent (SGD). The gradient descent is the partial derivative of the weight loss function.

Image convolution is the multiplication by elements of two matrices, followed by the sum of all elements of the cross-correlation of a subimage $w\left(x,\;y\right)$ (called the Kernel or mask) of size K×L over image $f\;\left(x,\;y\right)$ of size M×N, where K ≤ M and L ≤N. The correlation between $w\;\left(x,\;y\right)$ and $f\;\left(x,\;y\right)$ at a point $\left(i,\;j\right)$ can be defined by the formula:$$C\left(i,j\right)=\overset{i,-1}{\underset{x=0}{\sum }}\overset{K=1}{\underset{y=0}{\sum }}w\left(x,y\right)f\left(x+i,y+i\right)$$
where i=0,1,… M=−1; j=0,1,…,N−1, and the sum is related to the image area where w overlaps f.

Convolution can use different kernel patterns to generate linear, nonlinear, or morphological filters on the image. These filters enhance specific features of the image, such as non-directional edge enhancement. In this way, the application of convolutions using a variety of kernels enlarges the image dimension and highlights attributes in each of the new image channels. CNNs use the pattern of highlights arising from each filter in lower-level layers to detect higher-level objects in the image at deeper layers of the network.

This methodology allows the model to extract patterns that are distributed in the image and is the basis of image pattern recognition today. This application as a classification method has proven to be a useful tool to aid diagnosis in the interpretation of medical images. This model can help less experienced pathologists and can be used as a training and teaching tool for new professionals.

## 2. Methods

### 2.1. Database

We used two major publicly available databases, TCGA and METABRIC [31], as well as data from breast cancer patients treated at Baghdad Teaching Hospital (BTH) in Iraq between 2018 and 2021. The local ethics council had already approved the project for clinical data collection and image acquisition for the BTH cohort. The TCGA has clinical and histopathological data on 1098 breast cancer patients, including histology photos of all of them. METABRIC is a database that contains clinical and histological information on 1992 breast cancer cases, and 564 histology pictures are available. The TCGA data had already been obtained and were being used for the system’s first development. Clinical data from 1967 BTH cohort patients had already been obtained, and histological pictures will be digitally acquired in the near future. The histology pictures from METABRIC and BTH were used for model validation and survival prediction.

### 2.2. Development of the Proposed System

The proposed system is composed of four modules:(1)A digital image reading module.(2)A preprocessing, normalization, and image selection module.(3)A nuclear segmentation module.(4)A machine training module.

The entire system was developed in the Python programming language, and the goal is to offer the tool in a web system:(i)Digital image reading module: The images from the TCGA and METABRIC databases were acquired with the Aperio^®^ scanner, which generated an SVS file that contained four images in TIFF format with different sizes. The system used the OpenSlide library to read the images, and the OpenCV library was used to visualize the image (ROI) corresponding to the tumor ROI. The preprocessing and normalization system reads the PNG images from the database, allowing the user to determine changes in the image dimension. An image normalization module was implemented. We will use a tool developed by Macenko M et al., (2009) [32] that normalizes the staining intensity of the slides.(ii)Nuclear segmentation module: the nuclear segmentation strategies were implemented using Mask R-CNN after the construction of the definitive image bank, with the exclusion of low-quality images.(iii)Machine training module: This module is the foundation of machine learning. It was implemented after nuclear targeting. We intend to test different classic architectures, including AlexNet, VGGnet, Lenet, and GoogLeNet. The entire CNN will be developed with the Keras library, using Tensorflow as a learning tool.

AlexNet: AlexNet was used to analyze the histopathological pictures of breast cancer (BC) and classify mammograms into benign (normal) and malignant (abnormal) tumors. Then, the breast cancer characteristics from the histopathologic photos were extracted. AlexNet is a deep convolutional neural network architecture that was developed by Alex Krizhevsky, Ilya Sutskever, and Geoffrey Hinton in 2012. It was the first architecture to win the ImageNet Large Scale Visual Recognition Challenge (ILSVRC) with a top-5 error rate of 15.3%. The architecture consists of eight layers, including five convolutional layers and three fully connected layers. The first and second convolutional layers have receptive fields of 11 × 11 and strides of 4, while the remaining convolutional layers have receptive fields of 3 × 3 and strides of 1. The fully connected layers have 4096, 4096, and 1000 neurons, respectively.

VGGnet: By using VGGnet deep learning algorithms, the pictures were first segmented, then categorized according to their attributes, and finally molded. VGG is a convolutional neural network architecture that was developed by the Visual Geometry Group at the University of Oxford. The architecture consists of 16 or 19 layers and is known for its use of small convolutional filters (3 × 3) with strides of 1 and max pooling layers with strides of 2. This architecture was able to achieve a very high accuracy on the ImageNet dataset.

Lenet: Lenet excels in processing large-scale images because some of its artificial neurons may react to nearby cells within the coverage region. In addition, it is used for the automatic categorization of histopathological pictures of human cancer.

GoogLeNet: For the purpose of identifying and classifying cancerous cells in breast cytology pictures, GoogleLeNet’s architecture is described. Three well-known CNN architectures, GoogleLeNet, VGGNet, and ResNet, individually extract various low-level characteristics. Subsequently, for the classification job, merged features are supplied into a fully linked layer. GoogLeNet is a convolutional neural network architecture that was developed by Google in 2014. It was the winner of the ILSVRC 2014 competition with a top-5 error rate of 6.67%. The architecture is known for its use of inception modules, which are a combination of multiple convolutional filters and pooling layers. The inception modules allow the network to learn multiple scales of features at once, which helps to reduce the number of parameters in the network and improve performance. The architecture has a total of 22 layers and uses both 1 × 1 and 3 × 3 convolutional filters.

These models are all recent developments in the field of deep learning and computer vision. ResNet is a deep residual network that solves the vanishing gradient problem, making it easier to train deep neural networks. EfficientNet is an extension of ResNet that is optimized for the efficient use of computation resources. ResNeXt is an improvement on ResNet that introduces grouped convolutions to better capture the relationships between channels.

Comparing these models to the methods used in this study requires a deeper understanding of the specific methods used and the objectives of the study. However, it is likely that these models have improved performance and efficiency compared to the older methods. It is also likely that these models have different strengths and weaknesses and may be more or less suitable for specific tasks or scenarios.

Figure 3 illustrates the block diagram of the proposed methodology.

After the training process and the prediction model are established, we will apply and test the histological classification model as a predictor of survival in breast cancer. This project phase will be developed with collaborators from the University of Cambridge in the United Kingdom.

The entire neural network training phase was performed with the TCGA (The Cancer Genome Atlas) database, and the Molecular Taxonomy of Breast Cancer International Consortium (METABRIC) image database was used to test the classification and analyze its impact on the prediction of recurrence and death from the disease. METABRIC is a consortium between the University of Cambridge and the British Columbia Cancer Agency in Vancouver, Canada. The patients in this study had a long-term follow-up, all were submitted to a standardized treatment, and the histological classification was performed in the laboratories of both institutions, which makes the survival analysis and the histological classification quite consistent. We will apply the model trained with the images in the TCGA database for the histological classification of samples from the METABRIC and BTH studies. Later, we will analyze the impact of automated classification on survival compared to traditional classification.

## 3. Results

In this research, a binary classification model was developed to classify images based on their quality. A substantial proportion of the subimages that were generated were found to not exhibit sufficient tumor cell sampling or represent normal tissue, out of focus areas, or image artifacts. To address this issue, a convolutional neural network (CNN) was applied after filtering the images through an unsupervised K-means algorithm. The K-means algorithm classified each pixel of the image based on its similarity to other pixels, resulting in the formation of a mask that converted non-similar pixels to white. The proportion of non-white pixels was used to determine the representativeness of neoplastic cells in each image, allowing for a binary classification of images with good and bad quality before final training. The aim of this approach was to improve the efficiency of the training process by only considering high-quality images for the final model.

Following the image normalization methodology in H&E, separating each of the H and E channels was possible. With this approach, we were able to extract the most representative image features of the cell nucleus (the H channel) after subtracting the eosin stain. The K-means unsupervised clustering algorithm was applied to original images, normalized images, and the eosin-subtracted image (H channel). This algorithm performed the clustering of each pixel based on similarity and the number of clusters (Ks). We applied the method with k = 3 and k = 4 (Figure 4).

The second filtration system is under development. A total of 3597 images were rated as good or poor quality by a doctor. We are testing the performance of CNNs with different depths. The images were separated into subgroups for training and testing (40% for testing), and the images were reduced to 2562 × 1282 pixels. The system uses the ImageDataGenrator () class for data augmentation and we apply training with 50 and 100 epochs as a default.

Different optimization functions were tested, and Adam’s best performance was with an initial learning rate of 0.001. After some adjustments to the dropout layers, the model reached an accuracy of around 80%. However, evident overfitting could still be observed around the 20th epoch (Figure 5).

Note that, visually, the cell nucleus segmentation was more accurate when applied to the image with eosin subtraction (H channel).

Figure 6 shows that the process of analyzing medical images of breast cancer involves collecting a large dataset, dividing the images into two sets for ROI analysis and comparison, applying an ROI analysis to identify regions of interest, training two CNN models using filtered and unfiltered image sets, evaluating and comparing the models’ accuracies, and analyzing the results to identify strengths and weaknesses. The use of ROI analysis in a CNN model is believed to improve the medical image classification performance.

## 4. Discussion

Before executing the final training, it is crucial to classify the subimages according to their quality, as a substantial portion of the subimages (approximately 30%) lack adequate tumor cell representation or depict areas of normal tissue, out-of-focus regions, or image artifacts. To address this, we are developing a binary classification model to differentiate between images of good and poor quality. The approach utilizes a convolutional neural network (CNN) and a K-means algorithm, which classifies each pixel based on its similarity to other pixels. This allows the creation of a mask that turns all pixels other than blue-toned pixels in the hematoxylin to white, thus inferring the representativeness of the neoplastic cells in each image. Our method also employs H&E image normalization to separate the H and E channels and identify the most typical cell nucleus image properties. The subimages are grouped into smaller sets to facilitate training and evaluation, and the classification of malignant neoplasms is based on the degree of differentiation. This research utilizes artificial neural networks to develop an automated system for the histological grading of breast cancer, an important branch of pathology. The results of this study were compared to previous studies in terms of sample size, methods, results, and contributions, as presented in Table 1. Table 2 compares the proposed method to AlexNet, VGGnet, Lenet, and GoogLeNet. Table 3 displays the performance of the proposed method. The proposed model uses a binary classification approach to classify images based on their quality, which is necessary since a significant fraction of the images are not informative. The model also uses a normalization technique to separate the H and E channels and retrieve the most typical cell nucleus image properties. The model is based on convolutional neural networks, which have shown promise in previous studies for classifying medical images. The results of the proposed model were compared with state-of-the-art models such as AlexNet, VGGnet, Lenet, and GoogLeNet, and the proposed model showed improved performance in terms of the sample used, the methods, the results, and the contribution to the research.

This table compares five different neural network architectures, including AlexNet, VGG16, GoogLeNet, LetNet, and the proposed method, based on their salient features, number of parameters, and top-5 and top-1 accuracy.

The architecture with the largest number of parameters is AlexNet, at 62378344. However, it has a lower top-1 accuracy compared to the other architectures, at 63.30%. On the other hand, the architecture with the highest top-1 accuracy is the proposed method, at 79.00%.

VGG16 has a larger number of parameters compared to AlexNet and GoogLeNet, but it also has higher top-5 and top-1 accuracy, at 91.90% and 74.40%, respectively. GoogLeNet has a smaller number of parameters compared to VGG16, but it still has high top-5 and top-1 accuracy, at 92.20% and 74.80%, respectively.

LetNet has the smallest number of parameters among the five architectures, at 8,062,504, but it still has relatively high top-5 and top-1 accuracy, at 93.34% and 76.39%, respectively.

The proposed method has a slightly larger number of parameters compared to LetNet and a smaller number of parameters compared to AlexNet and VGG16. It also has higher top-5 and top-1 accuracy compared to LetNet and a top-5 accuracy comparable to VGG16.

## 5. Conclusions

Based on the partial results that have been obtained, we can state that (i) the use of image normalization methods followed by clustering segmentation techniques allows the identification of ROIs and the filtering of images with low representation of tumor cells; (ii) the use of convolutional networks for the selection of low-quality histological images is feasible; and (iii) there is a need for more robust data processing systems for the optimization and training of histological images. The following areas should be addressed in future research:Further testing and validation of the proposed method using a larger dataset of medical images to improve the robustness and generalizability of the results.Investigating the use of other deep learning techniques, such as deep neural networks (DNNs) or recurrent neural networks (RNNs), in combination with the CNNs for improved performance in identifying ROIs and classifying histological breast cancer subtypes.Developing a more efficient and automated method for selecting and preprocessing the medical images to be used in the CNNs in order to reduce the need for manual input and improve the speed and accuracy of the system.Investigating the use of different image normalization and clustering segmentation techniques to improve the performance of the system in identifying ROIs and filtering images with low representation of tumor cells.Developing a user-friendly and low-cost AI model that can be used by healthcare professionals in remote areas, with the goal of improving access to accurate and reliable breast cancer screening and diagnosis.

## Figures and Tables

**Figure 1 medicina-59-00487-f001:**
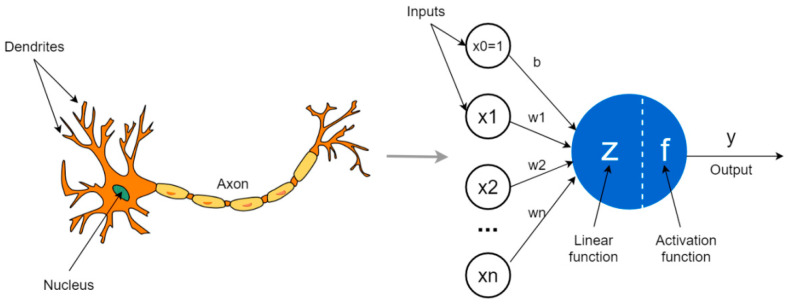
Schematic model of the perceptron and its similarity to the structure of a neuron [24].

**Figure 2 medicina-59-00487-f002:**
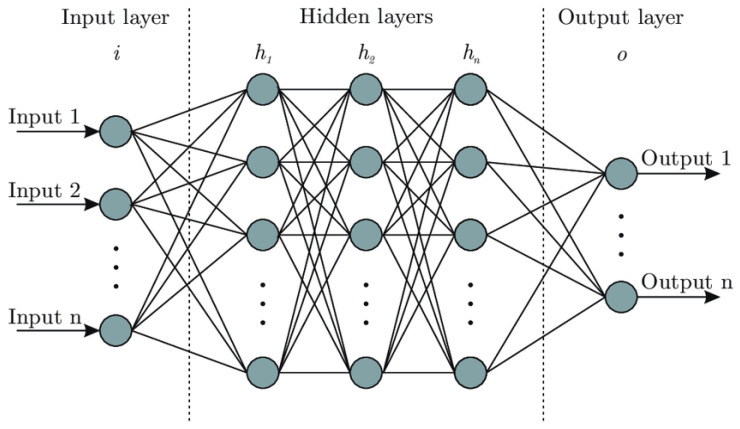
Architecture of an artificial neural network (MLP) [30].

**Figure 3 medicina-59-00487-f003:**
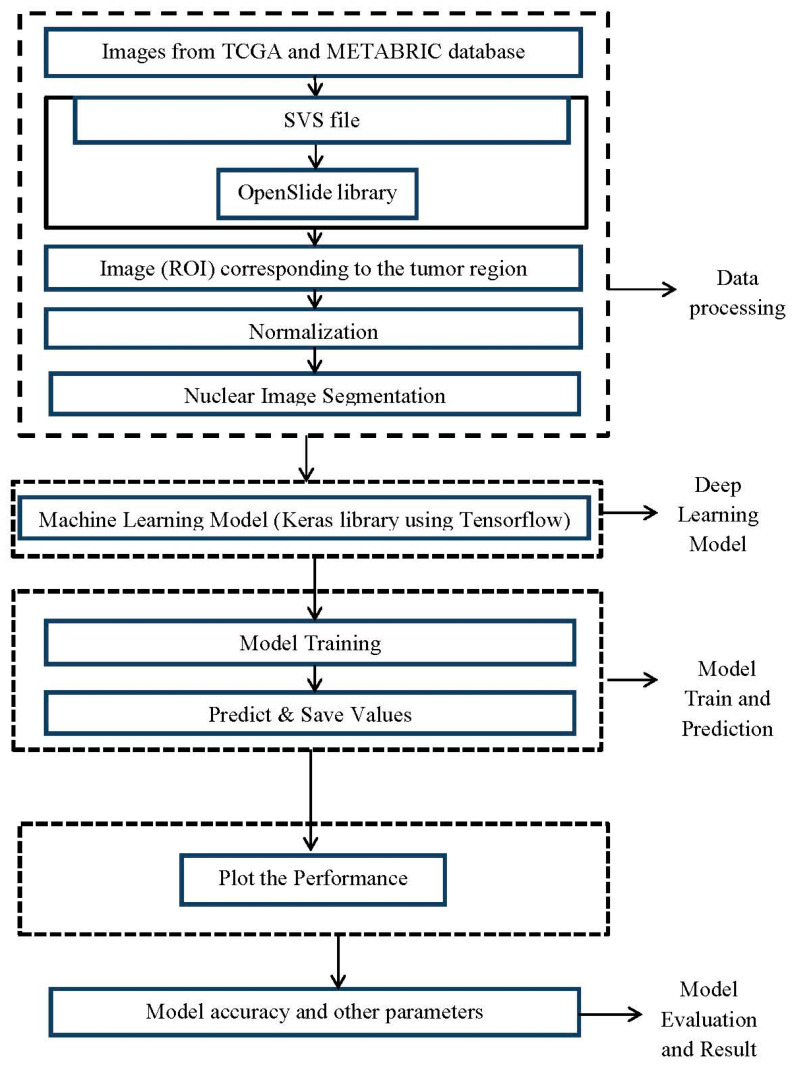
Block diagram of the proposed methodology.

**Figure 4 medicina-59-00487-f004:**
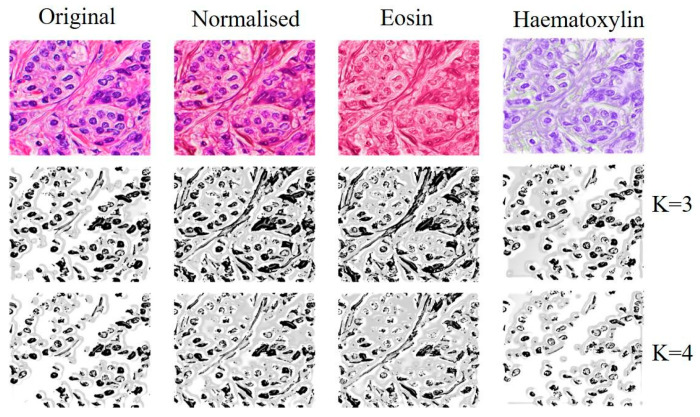
Segmentation using the K-means algorithm with k = 3 and k = 4.

**Figure 5 medicina-59-00487-f005:**
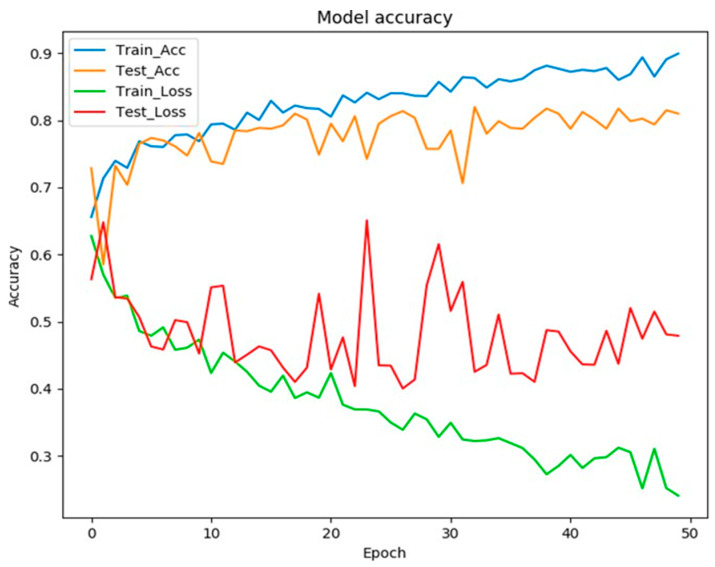
Accuracy and error during CNN training to predict high- and low-quality images. Note that around the 20th epoch, the accuracy of the model in the test group continued to rise, and at validation, it stabilized. The error curve also did not follow the decline.

**Figure 6 medicina-59-00487-f006:**
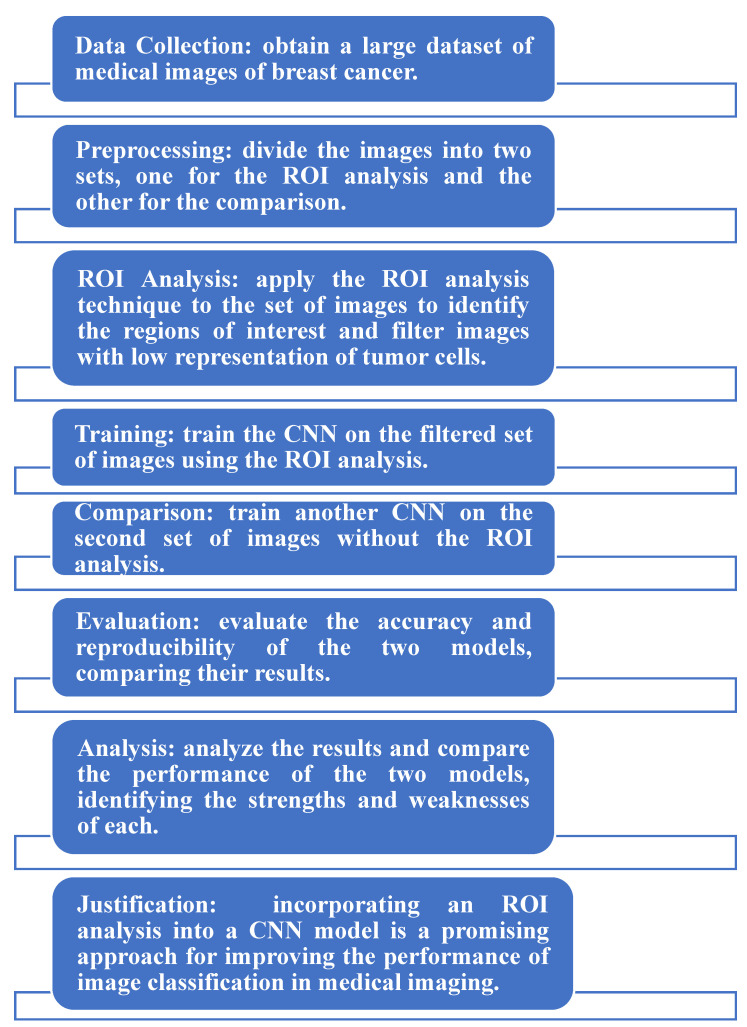
The CNN model with ROI and without ROI.

**Table 1 medicina-59-00487-t001:** Summary of results of previous studies.

Article Citation	Sample and Methodology	Results	Contribution of Study
S Albarqouni. et al., (2016) [23]	**S:** They provided an experimental investigation on crowdsourcing learning. Annot8, a self-implemented web platform based on the Crowdflower API, was used in the experimental setting to realize picture annotation tasks for a publicly accessible biomedical image database. **M:** 1. Multi-scale CNN model; 2. Layer of aggregation (AG).	The findings demonstrated the importance of data aggregation integration and provided useful insights into how deep CNN learning using crowd annotations works.	They developed a fresh idea for crowdsourcing learning in this study. Through an extra crowdsourcing layer, the novel multi-scale CNN AggNet was made to handle data aggregation directly as part of the learning process.
Y Li et al., (2019) [24]	**S:** The diagnosis of breast cancer histology images with hematoxylin and eosin staining was non-trivial. Two pathologists labeled images as normal, benign, in situ carcinoma, or invasive carcinoma according to the predominant cancer type in each image, without specifying the area of interest. **M:** 1. The framework; 2. Sampling patches; 3. Feature extractor; 4. Screening patches; 5. Image-wise classification.	As a consequence, when the method described in this study was used to categorize breast cancer histology photos into four classes, it achieved 95% accuracy on the first test set and 88.89% accuracy on the whole test set.	For the classification of breast cancer histology images, they suggested patch sampling utilizing the multi-Size and discriminative patches approach. Two types of patches of various sizes were extracted, each of which included characteristics at the cellular and tissue levels. They created a classification framework that extracted features from the patches using feature extractors and computed the final feature of each whole image for classification through a classifier. They designed a patch selecting method to select more discriminative patches based on the CNN and K-means.
YS Vang et al., (2018) [33]	**S:** scanning the breast using an X-ray to check for changes in the United States **M:** In order to categorize patches, they first proposed utilizing Inception V3 to classify patches at the patch level. These extracted features were then transferred to a second layer of ensemble prediction fusion using GBM, logistic regression, and a support vector machine (SVM) to improve the predictions. The forecasts were made at the patch level.	The framework achieved an accuracy score of 87.5%, a 12.5% improvement over the baseline score. In spite of the refinement model, the model gave a 6% improvement over the standard model. For the normal, benign, in situ, and invasive classifications, each attained sensitivities of 77.8%, 66.7%, 88.9%, and 88.9%. This supports the inclusion of a binary class refinement step only for the benign and normal classes.	The experimental findings revealed that the approach outperformed the leading model by 12.5%.
Jebarani, et al., (2021) [34]	**S:** A digital dataset for screening mammography (DDSM) was received from the University of South Florida. **M:** An adaptive median filter was used for noise reduction, picture quality improvement, edge preservation, and smoothing. The multi-variant analysis and prediction rate for the suggested technique were determined using the ANOVA test. A hybrid mix of segmentation and detection was used on breast cancer.	The suggested approach enabled binary outcomes to indicate whether the tissue was benign, normal, or malignant when a mammographic picture included microcalcifications.	Early diagnosis improves treatment choices and saves lives. The versatile methodology combining modern segmentation methodologies with machine learning techniques achieved this goal by suggesting a new parameter for assessing the performance of K-means and a Gaussian mixture model (GMM).
Nomani. et al., (2022) [35]	**S:** In this study, 64 samples from the benign class and 51 samples from the malignant class from the MIAS dataset were evaluated. **M:** artificial intelligence, image processing, and CNN	1. In 98.8% of the instances, 905 pictures with various diseases were appropriately identified. 2. The specificity of PSOWNNs was 98.8%. 3. Since PSOWNNs had a 98.6% accuracy rate, 830 people were appropriately identified as having breast cancer. 4. The results demonstrated that a CNN can gain additional functionalities that are more beneficial than methods that do not account for this. 5. The CNN showed that 799 people (94.9%) correctly identified 905 breast cancer patients. 6. In total, 106 pictures had incorrect diagnoses.	This article’s goal was to examine several methods for spotting breast cancer using artificial intelligence and image processing. The findings showed that the strategy described in this study may provide an excellent fusion picture that is superior to that produced by certain advanced image fusion algorithms in terms of both visual and objective evaluations.

**Table 2 medicina-59-00487-t002:** Comparison among various methods.

Architecture	Salient Feature	Number of Parameters	Top-5 Accuracy	Top-1 Accuracy
AlexNet	Deeper	62,378,344	84.60	63.30
VGG16	Fixed-size kernels	138,357,544	91.90	74.40
GoogLeNet	Fixed-size kernels	23,000,000	92.20	74.80
LetNet	Shortcut connections	8,062,504	93.34	76.39
Proposed method	Wider parallel kernels	22,910,480	94.50	79.00

**Table 3 medicina-59-00487-t003:** Performance evaluation of the proposed method.

Proposed Network	Accuracy (Epoch)	Training Time
Pretrain (160 × 160)	69.05 (45)	70.19 h
Target (224 × 224)	72.61 (49)	178.45 h
Resized Epoch 10	72.91 (46)	146.72 h
Resized Epoch 30	72.79 (46)	114.23 h
Resized Epoch 50	71.36 (45)	109.81 h

## Data Availability

Not applicable.

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
