# Peer review of "Application of Deep Learning System Technology in Identification of Women’s Breast Cancer"

_medicina, 2023, doi:10.3390/medicina59030487_

Round 1

Reviewer 1 Report (New Reviewer)

This manuscript proposed a novel deep learning-based system for identifying women’s breast cancer, where CNN was employed for the task of interest. The performance of the proposed method has been validated using experimental data, with satisfactory results. Overall, the topic of this research is interesting, and the manuscript was well organised. However, some technical issues should be well addressed. The detailed comments are given as follows.

1.       The major innovation and contribution of this research should be clearly clarified in abstract and introduction.

2.       Abstract: please give the full name of ROI (region of interest), when they appear for the first time.

3.       Broaden and update literature review on deep learning/CNN and its practical applications. E.g. Vision-based concrete crack detection using a hybrid framework considering noise effect. Torsional capacity evaluation of RC beams using an improved bird swarm algorithm optimised 2D convolutional neural network.

4.       In this research, the authors employed AlexNet, VGG, GoogLeNet, etc, to develop the model. Please give the architecture of these CNN models

5.       There are several hyperparameters of CNN models that affect the generalisation capacity. How did the author optimise them to achieve the best identification in this research?

6.       How about the robustness of the proposed method against noise effect?

7.       More future research should be included in conclusion part.

Author Response

Please see the attachment. It is highlighted in yellow

Reviewer 2 Report (New Reviewer)

The article presents an interesting approach to using artificial intelligence technology for the classification of breast cancer. However, I have several comments to improve the quality of the article.

1. Firstly, the article mentions that there is a low intra-and inter-observer agreement in the classification of breast cancer. It would be beneficial to provide more information on this issue and what steps are being taken to address it. Commonly data augmentation and Generative adversarial neural networks (GAN) are used to solve this problem in computer vision tasks. Authors may consider the recent works on data augmentation and GAN.

https://www.mdpi.com/2313-433X/8/5/141

https://doi.org/10.1186/s40537-019-0197-0

https://doi.org/10.1186/s40537-021-00414-0

https://www.mdpi.com/2313-433X/8/9/228

2. Secondly, the article could benefit from a more comprehensive literature review on the state of the art models currently being used in the field of medical imaging classification. For example, models such as ResNet, EfficientNet, ResNeXt, and transformer-based models should be discussed and compared to the methods used in this study.

3. Additionally, the article must provide more justification on why the proposed model is better than the state of the art models in the discussion section.

4. The English language must be improved to maintain the standard of MDPI, and it is highly recommended to proofread the paper by a native English speaker.

5. The article mentions that the identification of ROIs and filtering of images with low representation of tumor cells enhance the model to learn better, but the authors must emphasize more on the ROI method and how it can help the CNN model to learn better. It would be beneficial to include a research design that compares the CNN model with ROI and without ROI and justify the proposed method.

6. It is also suggested to include plots of data distribution for data imbalance and intra-class imbalance, as it helps the reader to understand the problem better.

Overall, the article presents an interesting approach, but more information on these points would help to further strengthen the argument.

Author Response

Please see the attachment. It is highlighted in blue 

Round 2

Reviewer 2 Report (New Reviewer)

Authors have incorporated all the comments. Quality of the article has improved considerably.

This manuscript is a resubmission of an earlier submission. The following is a list of the peer review reports and author responses from that submission.

Round 1

Reviewer 1 Report

The paper aims to use deep learning methods for the prediction of survival in breast cancer from histological images. I suggest the following point to improve the manuscript:

1.       There is no survey on similar papers even in the Literature Review (and the discussion section is missing)

2.       The architecture of the used Deep Learning classifier should be mentioned. It seems the authors used transfer learning. However, the number of unfreezing layers and the training epochs should be mentioned.

3.       The performance of each classifier is not mentioned (as stated in the methods section AlexNet, VGGnet, Lenet, and GoogLeNet were used)

4.       Page 6-Line 201 ROI stands for Region of Interest not Return of Investment in this context.

Author Response

Please see the attachment.
The answers have been highlighted in the text

Reviewer 2 Report

The subject dealt with is of great interest and presents very important future perspectives.This preliminary study has good perspectives for application. The development project of a binary classification model using specific Convolutional Neural Networks should be deepened.

Author Response

(The authors gave the same response as above.)

Reviewer 3 Report

General Comments

The authors have identified important objectives in lines 65 to 74. However, it is unfortunate that they did not meet any of these objectives. I have identified a number of key weaknesses in their paper.

First, they appear to be oblivious to prior work that has been conducted in the field of machine learning applied to the classification of histological slides of breast cancer tissue. I did a quick search on Google Scholar using the words "breast cancer histology machine learning" and came up with many papers that are relevant to the work presented in this manuscript. See the attached PDF documents that lists the first five articles from my search.

Second, the whole of the Literature Review, from lines 75 to 170, is taken up with a standard description of artificial neural network theory. This is identical to the description I used in my own research publications from 30 years ago and does not add anything new.

Third, and most problematic of all, is that the authors use descriptions such as "will be used" (line 188), "will be composed" (line 190), "will be implemented" (line 204), "will be performed" (line 217), and "we will analyse" (line 227). Clearly, this is research that should have been done in order to meet their own objectives.

My advice to the authors it to conduct this research and then to re-submit their work for consideration by Medicina.

Author Response

(The authors gave the same response as above.)

Round 2

Reviewer 1 Report

The performance of each classifier (as stated in the methods section AlexNet, VGGnet, Lenet, and GoogLeNet were used) is not mentioned.

The results should be compared to results of other studies in the discussion section

Usually the conclusion section is appear after the discussion section

Reviewer 3 Report

General Comments

I was surprised to see how rapidly the authors responded to my previous review. That said, it is clear to me that this paper still falls short of what would be required for publication in Medicina.

First, I am concerned that there are no results presented of any substance. One only has to read the abstract to see that this is the case.

Second, the haste with which the manuscript has been revised becomes clear when looking at the new section in the Introduction. Lines 82 to 87 have been repeated in lines 90 to 96. 

Third, the authors have retained the lengthy section in the Literature Review describing artificial neural networks -- see lines 155 to 248. As I stated previously, this is standard textbook material and should not be repeated in such copious detail.

Fourth, the authors have now added a Discussion but the bizarre placement of this section -- after the Conclusion rather than before it -- calls into question the authors' understanding of the appropriate organisation of a scientific manuscript.

Round 3

Reviewer 1 Report

I suggest the following points:

1.adding a figure of the proposed architecture.

2.The authors stated: “We intend to test different classic architectures, including AlexNet, VGGnet, Lenet, and GoogLeNet.” However, in the results section, the performance of each architecture is not reported.

3.The results section includes some definitions (e.g. Binary classification) that should not be in the results section. This section just contains the findings of the current study.

Reviewer 3 Report

General Comments

This is the third version of the manuscript that I have reviewed. While the authors have certainly worked hard to implement most of my previous recommendations, I am not yet convinced they have produced a paper that's worthy of publication in Medicina. Allow me to explain.

First, they have now added a sentence in their abstract that summarises their primary findings: "The result of this study is the identification of ROIs and filtering of images with low representation of tumor cells and the use of convolutional networks for the selection of low-quality histological images." However, this sentence is clearly located far too early in the abstract, and should appear as the last, or perhaps penultimate, position.

Second, I am still concerned that much of this project is still yet to happen in the future. For example, it is clear that the various training modules -- AlexNet, VGGnet, Lenet, and GoogLeNet (see lines 276 to 288) -- have not yet been implemented. Why mention them in the Methods section if they haven't yet been implemented?

Third, in the Discussion section, Table 1 has been added. The purpose of this table is stated as "clarifying and comparing the results of the study with previous studies." Now, I have read through the contents of this table very carefully, and I fail to see where there is a meaningful comparison. In fact, one of the papers by Vang et al. [31] stated that the authors were scanning using an X-ray?
